# Risk Factors for Pancreatic Cancer in Patients with New-Onset Diabetes: A Systematic Review and Meta-Analysis

**DOI:** 10.3390/cancers14194684

**Published:** 2022-09-26

**Authors:** Claudia Mellenthin, Vasile Daniel Balaban, Ana Dugic, Stephane Cullati

**Affiliations:** 1Department of Surgery, Hospital Fribourg, 1700 Fribourg, Switzerland; 2Faculty of Science and Medicine, Institute of Family Medicine, University of Fribourg, 1700 Fribourg, Switzerland; 3Arztpraxis am Bager, 3185 Schmitten, Switzerland; 4Faculty of Medicine, Carol Davila University of Medicine and Pharmacy, 050474 Bucharest, Romania; 5Department of Gastroenterology, Central Military Emergency University Hospital, 010825 Bucharest, Romania; 6Department of Gastroenterology, Friedrich-Alexander-Universität Erlangen-Nürnberg (FAU), Medizincampus Oberfranken, 95445 Bayreuth, Germany; 7Department of Medicine, Huddinge, Karolinska Institutet, 14186 Stockholm, Sweden; 8Population Health Laboratory, University of Fribourg, 1700 Fribourg, Switzerland; 9Department of Readaptation and Geriatrics, University of Geneva, 1211 Geneva, Switzerland

**Keywords:** pancreatic cancer, new onset diabetes, cancer screening, risk factors, meta-analysis

## Abstract

**Simple Summary:**

New onset diabetes patients are a high-risk group for pancreatic cancer. Since pancreatic cancer is responsible for less than 1% of new-onset diabetes cases, testing all of them might lead to an unfavorable risk/benefit balance. Additional risk factors can contribute to a better definition of the population that needs further screening. Currently, 22 studies examining additional risk factors have been published, but often they have a limited number of participants for the individual risk factor. By pooling their results in a meta-analysis, we could establish the magnitude of several risk factors. We found that pancreatic cancer cases were older than controls by 6.14 years (CI 3.64–8.65, 11 studies). Among new-onset diabetes patients, the highest risk of pancreatic cancer involved a family history of pancreatic cancer (3.78, CI 2.03–7.05, 4 studies), pancreatitis (5.66, CI 2.75–11.66, 9 studies), gallstones (2.5, CI 1.4–4.45, 4 studies), weight loss (2.49, CI 1.47–4.22, 4 studies), and high/rapidly increasing glycemia (2.33, CI 1.85–2.95, 4 studies) leading to more insulin use (4.91, CI 1.62–14.86, 5 studies). Risk factors or symptoms were distinct in the new-onset diabetes patient group. They are strongly connected to pancreatic cancer and are ideal for targeted screening, using a score or model as the first step.

**Abstract:**

(1) Background: Patients with new-onset diabetes (NOD) are at risk of pancreatic ductal adenocarcinoma (PDAC), but the most relevant additional risk factors and clinical characteristics are not well established. (2) Objectives: To compare the risk for PDAC in NOD patients to persons without diabetes. Identify risk factors of PDAC among NOD patients. (3) Methods: Medline, Embase, and Google Scholar were last searched in June 2022 for observational studies on NOD patients and assessing risk factors for developing PDAC. Data were extracted, and Meta-Analysis was performed. Pooled effect sizes with 95% confidence intervals (CI) were estimated with DerSimonian & Laird random effects models. (4) Findings: Twenty-two studies were included, and 576,210 patients with NOD contributed to the analysis, of which 3560 had PDAC. PDAC cases were older than controls by 6.14 years (CI 3.64–8.65, 11 studies). The highest risk of PDAC involved a family history of PDAC (3.78, CI 2.03–7.05, 4 studies), pancreatitis (5.66, CI 2.75–11.66, 9 studies), cholecystitis (2.5, CI 1.4–4.45, 4 studies), weight loss (2.49, CI 1.47–4.22, 4 studies), and high/rapidly increasing glycemia (2.33, CI 1.85–2.95, 4 studies) leading to more insulin use (4.91, CI 1.62–14.86, 5 studies). Smoking (ES 1.20, CI 1.03–1.41, 9 studies) and alcohol (ES 1.23, CI 1.09–1.38, 9 studies) have a smaller effect. (5) Conclusion: Important risk factors for PDAC among NOD patients are age, family history, and gallstones/pancreatitis. Symptoms are weight loss and rapid increase in glycemia. The identified risk factors could be used to develop a diagnostic model to screen NOD patients.

## 1. Introduction

The incidence of pancreatic ductal adenocarcinoma (PDAC) doubled over the last 2 decades [1]. The cumulative lifetime risk is 0.91% [2]. Diagnosis of PDAC comes too late for curative treatment in 80% of cases. This contributes to PDAC being one of the deadliest cancers worldwide, accounting for 4.7% of all cancer-related deaths [3]. Among diagnosed patients, the 5-year survival rate does not exceed 10% [4]. In countries that have screening programs for breast and colorectal cancers, PDAC has become the second most frequent cause of cancer mortality [5].

It has been established that all cancers discovered in the first years after diabetes diagnosis were already present and caused the diabetes, and several underlying mechanisms are under research [6,7,8,9,10,11,12]. Diabetes or prediabetes is often the first symptom of PDAC: diabetes diagnosis happens up to 3 years before the cancer diagnosis [13]. Among pancreatic cancer patients, about 80% have a diagnosis of either hyperglycemia or diabetes. Blood glucose levels slowly increase as early as 10 years before PDAC diagnosis, in the prediabetes range [14]. This has led to the idea that NOD or even prediabetes could be a potential clue to the early diagnosis of pancreatic cancer [15].

As pancreatic cancer is responsible for less than 1% of NOD cases, using a biomarker test for every patient with NOD might lead to an unfavorable risk/benefit balance if the performance of the test is not exceptional [16] (Figure 1).

To further stratify the group that would need biomarker and then imaging testing, the use of a simple model or score is interesting. This strategy of 3 sieves would be more cost-effective and cause less harm than a strategy leaning on biomarkers and imaging alone.

Currently, 22 studies examining additional risk factors have been published, but often they have a limited number of participants for the individual risk factor. Pooling their results in a meta-analysis should increase the precision.

Based on a systematic review with meta-analysis, this paper aims to assess PDAC risk in NOD individuals and to identify risk factors among NOD patients, which are needed for a stepwise diagnostic strategy.

## 2. Materials and Methods

We performed a systematic literature search and last updated it in June 2022 in the three major databases, PubMed (RRID:SCR_004846), Embase (RRID:SCR_001650), and Google Scholar (RRID:SCR_008878), using the terms described in Appendix A. We did not apply any search restrictions. The study is registered in the inplasy study registry (INPLASY202220065).

We included observational studies (both cohorts and case-control studies) reporting on NOD patients and assessing additional factors regarding the risk of developing PDAC. Our objectives were to identify these risk factors that further enrich the NOD population in PDAC occurrence. Also, we aimed to analyze the risk of PDAC in NOD patients compared to non-diabetic persons.

We excluded studies with the sole focus on biomarkers or medication. We did not include case reports, small case series, reviews, opinions, or articles without an English abstract. When we found interesting conference abstracts, we searched with the author's names for follow-up publications, and, if relevant, included those. As the data was presented in a very heterogenous way, we sometimes contacted the authors for additional data to be included in their study. However, not all authors answered (Appendix A, Table A1 of studies excluded at the full-text screening).

Two team members voted blindly during each step of the paper selection and quality assessment and made consensus decisions, resolving conflicts by discussion.

We extracted the following data from eligible studies: the name of the first author, journal and publication year, country and period, sample size, study type, patient characteristics, NOD definition, risk of PDAC in the NOD population, and additional risk factors (Figure 2).

### Data Analysis

For identifying studies and excluding duplicates, we used Covidence software (Veritas Health Innovation, Melbourne, Australia; RRID:SCR_016484), following the updated PRISMA 2020 guideline [17].

Studies reporting associations were used in the meta-analysis. Using the method of DerSimonian & Laird (an estimate of heterogeneity after the Mantel-Haentzel model), we performed a random-effects meta-analysis of risk factors that were reported in at least 3 papers either with a Risk Ratio or an Odds Ratio or with raw numbers that allowed us to calculate the Odds Ratio. All Confidence Intervals (CI) are 95%. All analyses were performed with STATA, version 16.1 (StataCorp, College Station, TX, USA).

First, the authors (C.M. & V.D.B.) performed a quality assessment using 10 criteria as defined in the paper by Hoy et al. in a specific bias assessment tool for prevalence studies [18]. We judged overall bias for selected papers, following the corresponding bias flags among the 10 criteria. As the overall number of studies per risk factor was small, we did not exclude any study. To determine the risk of publication bias, we used a funnel plot and the Egger test (Appendix A, Figure A2).

We extracted data on the definition of NOD/subgroups of duration, age, sex, ethnicity, lifelong smoking, alcohol abuse, family history of PDAC, gall stones/cholecystitis, pancreatitis, a rapid increase of glycemia, weight loss, insulin use, obesity, and hyperlipidemia. When more than 2 groups were reported, we combined groups, for example, former smokers + current smokers = lifelong smokers. Or introduced the most meaningful cut-off; for example, for groups of BMI (Body Mass Index) reported, we distinguished BMI < 30 = not obese, BMI ≥ 30 obese (Details in Appendix A).

We also extracted the percentage of NOD patients that developed PDAC (in the cohort studies) and the OR for PDAC for NOD versus no diabetes in the case-control studies.

## 3. Results

### 3.1. Studies

The search yielded 779 references, which we imported into Covidence. After removing duplicates and excluding irrelevant studies, we selected 15 studies for data extraction. Reference lists and citation searches (for studies that cited those we had already included) provided an additional 6 studies to be included in the analysis. There was one paper from other sources. Twenty-two studies were included. In total, 576,210 patients with NOD contributed to the analysis, of which 3560 had PDAC (Figure 2).

The study designs were heterogeneous, including retrospective cohorts (some with prospective analysis) (*n* = 13), case-control studies (*n* = 8), and one small prospectively recruited screening study, with recruitment at a diabetes clinic [19] (Table 1).

### 3.2. Risk Factors for PDAC in NOD Patients

The strongest demographic risk factor was older age. The overall mean age difference in the studies was more than 6 years (pooled age mean difference 6.14 years, CI 3.64–8.65, I^2^ = 96%, 11 studies), which seemed to be even more pronounced in European studies. Sex was not a statistically significant risk factor: the overall effect size (ES, either from odds ratio or incidence rate ratio) in overall studies was 1.07 for the male gender (CI 0.96–1.18, I^2^ = 28.6%, 18 studies). Race was analyzed in only a few studies, which showed that whites had a slightly higher risk for PDAC (ES 1.46, CI 1.25–1.71, I^2^ = 0.0%, 5 studies) (Figure 3).

Concerning other risk factors, smoking was a just barely statistically significant risk factor (ES 1.20, CI 1.03–1.41, I^2^ = 44.0%, 9 studies); the same was true for alcohol (ES 1.23, CI 1.09–1.38, I^2^ = 5.9%, 9 studies). Pancreatitis (ES 5.66, CI 2.75–11.66, I^2^ = 85.5%, 9 studies) and gall stones/cholecystitis (ES 2.5, CI 1.4–4.45, I^2^ = 87.0%, 4 studies) showed an increased risk. Positive family history of pancreatic cancer was a very strong risk factor, with an effect size of 3.78 (CI 2.03–7.05, I^2^ = 68.6%, 4 studies). Obesity (defined as BMI ≥ 30) was not associated with more pancreatic cancer cases within the studied populations of NOD (ES 0.67, CI 0.45–1, I^2^ = 84.3%, 5 studies).

Weight loss was a significant symptom, with an effect size of 4 (CI 3.1–4.9, I^2^ = 89.3%, 4 studies). A rapid increase in glycemia was significant in 7 studies [22,27,28,35,36,41,42], but it was reported with such heterogeneity that a meta-analysis was impossible for all studies (ES 2.33, CI 1.85–2.95, I^2^ = 6.7%, 4 studies). The rapid increase in glycemia could lead to more insulin use (ES 4.91 CI 1.62–14.86 I^2^ = 91.9%, 5 studies) in pancreatic cancer patients. A few studies showed a negative association with high blood lipids [19,25,26,38,42] (Figure 4).

### 3.3. Association between NOD and PDAC

All studies identified a strong association between NOD and PDAC. The overall effect size was 3.35 (CI 2.75–4.09, I^2^ = 83.3%), with a clear tendency of the ES to be higher when the interval since NOD diagnosis was shorter: in the first year after diabetes diagnosis, it was 5.52 (CI 3.61–8.46, I^2^ = 85.6%).

### 3.4. Proportion of NOD Caused by PDAC

As we assumed that all PDAC was present before the diabetes diagnosis and the cause of NOD, we calculated the cumulative percentage of observed PDAC diagnosis. It ranged from 0.13% in the Taiwanese registry study by Tseng et al. [25] to 2.7% in the prospectively recruited screening study by Illés et al. [19]. Studies excluding people under 50 found 0.74% (CI 0.63–0.85%) of PDAC cases among NOD patients. The overall cumulative percentage of PDAC in NOD patients was 0.36% (CI 0.3–0.42, I^2^ = 86.3%, 14 studies) (Figure 5).

## 4. Discussion

### Limitations and Strengths of the Study

Despite our systematic approach, we could have overlooked critical studies through our choice of search terms. We minimized this by using several formulations and searching references regarding the included papers. The most significant limitations of our findings are biases in the included studies and the disparity of representation of geographical regions. Many studies are from the USA, Europe, and Asia, and one is from Australia, but we could not identify any South American or African studies.

In extracting the data, we were limited by the heterogeneity of the included studies. To have enough data to analyze, we included studies with slightly different definitions. Definitions of how we extracted the data are in Appendix A. The results of our meta-analyses still show considerate heterogeneity, partially explained by the difference in inclusion criteria (for example, age), ethnicity, and definition of new-onset diabetes, all of which we also examined as risk factors or subgroups. Some risk factors might also interact with each other.

A strength of our review is that it gives a complete, systematic overview of the current body of evidence regarding additional risk factors for PDAC in NOD populations. Our paper is, to our knowledge, also the first to conduct a meta-analysis on the risk factors.

## 5. Conclusions

### 5.1. Interpretation of Findings

The association between diabetes and PDAC has long been recognized. Several papers have shown that the risk is highest directly after diagnosis and then decreases over subsequent years [41]. The association might be confounded by commonly shared risk factors such as obesity or chronic pancreatitis. The actual frequency of pancreatic cancer in the population of NOD is still unclear, as most studies are retrospective, and the percentage in the only prospective study is much higher. Currently, four prospective studies are recruiting patients and might bring more clarity [42,43,44,45].

It is essential to look specifically at the group of NOD patients, as they differ from the general population. For instance, NOD patients tend to be more obese than the general population, as obesity is a very important risk factor for diabetes mellitus. Within the population of NOD patients, obesity is not associated with more PDAC cases, as our analysis shows. In fact, the mean BMI of pancreatic cancer cases was lower than that of NOD controls. This might be even more pronounced through tumor-induced recent weight loss. It was surprising to find at most a weak association of smoking and alcohol abuse in this meta-analysis. Possibly these risk factors are more important for non-diabetic PDAC patients, or their importance has generally been overestimated.

Risk factors or symptoms that are distinct in the NOD patient group and are strongly connected to pancreatic cancer are ideal for targeted testing. They can be used for statistical model fitting. Our analysis showed that age, family history of PDAC, pancreatitis/cholecystitis, weight loss, and rapid increase in glycemia/necessity of insulin are robust candidates. A tendency to lower lipids, unusual in newly diagnosed diabetes patients, is also interesting. Unfortunately, some of the strongest risk factors are rather rare, which negatively impacts the sensitivity of such models. The correct balance between the frequency and magnitude of those risk factors remains to be found.

### 5.2. Importance of the Presented Work and Future Directions for Early Diagnosis Programs

Screening programs aim to diagnose cancer in the asymptomatic, early stages amenable to curative treatment. Scrutiny regarding balancing benefits and burdens, cost, survival extension, and quality-life years gain is essential. As pancreatic cancer has a low incidence in the total population, this is a challenge. The main risk of pancreatic cancer screening is a too-high rate of false-positive results, leading to unnecessary further investigations. Including the identified additional risk factors or symptoms can help define the target population.

A stepwise approach of first identifying a group with increased risk of pancreatic cancer within the NOD population through a scoring or diagnostic model and then further reducing the number of patients needing imaging by a biomarker test has been proposed by Pannala et al. [15]. Several studies have proposed scores to identify the best group for testing [19,22,35,36]. A scoring system has advantages, as it is objective and can be validated. Nevertheless, it also has disadvantages, such as being time-consuming for the family physician or challenging to apply when data is missing. The complexity of a scoring model should consider the balance between the accuracy of prediction and the simplicity of daily use. Considering the slightly different associations of risk factors in different regions (for example, the USA, Europe, Asia), such scoring might differ depending on the location. These regional differences are related not only to the characteristics of PDAC patients but also to NOD. Diabetes is closely related to diet and obesity, which are subject to socio-cultural and genetic influences. In the USA, the average age for diabetes diagnosis is lower than it is in Europe. In Asia, patients with a much lower BMI than that in western countries suffer from an increased risk for diabetes [46]. In conclusion, before using a score as a diagnostic model in a new population, it will need adaptation, or at least calibration and validation.

## Figures and Tables

**Figure 1 cancers-14-04684-f001:**
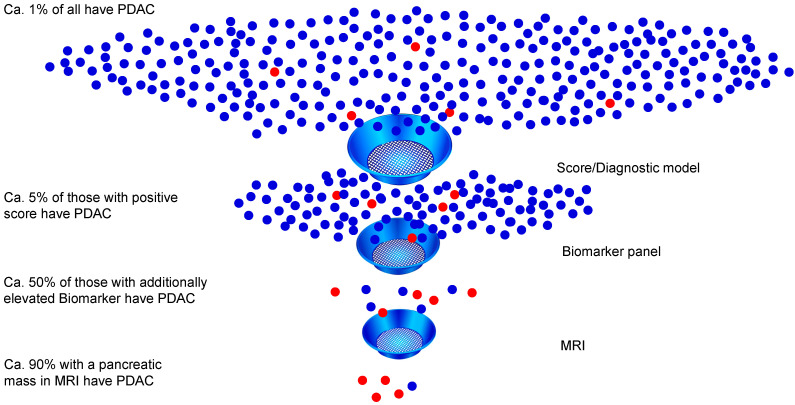
New onset diabetes patients—how to find those with pancreatic cancer.

**Figure 2 cancers-14-04684-f002:**
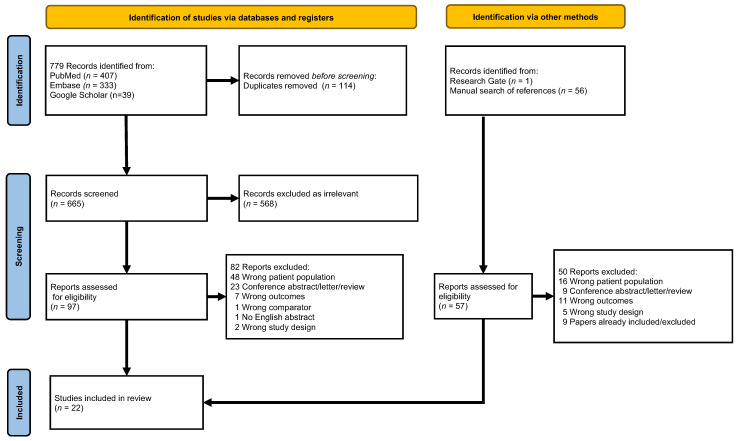
PRISMA flow diagram of the literature search and study selection process.

**Figure 3 cancers-14-04684-f003:**
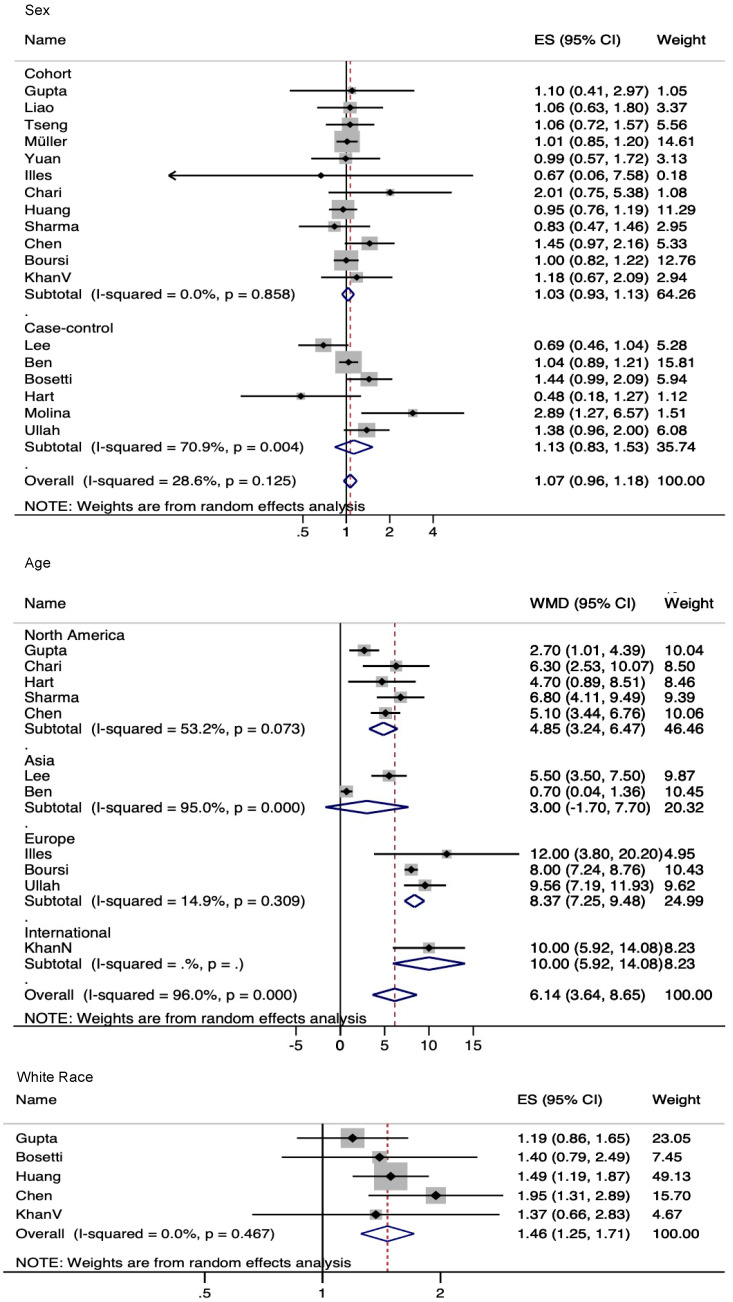
Meta-analysis of demographic risk factors.

**Figure 4 cancers-14-04684-f004:**
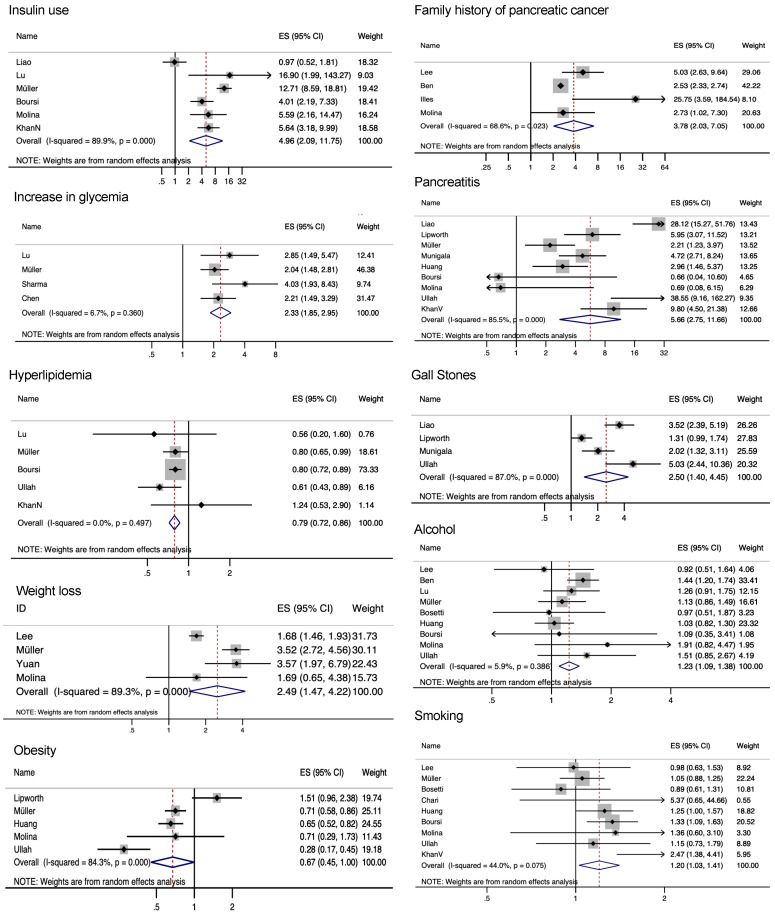
Meta-analysis of other risk factors/symptoms.

**Figure 5 cancers-14-04684-f005:**
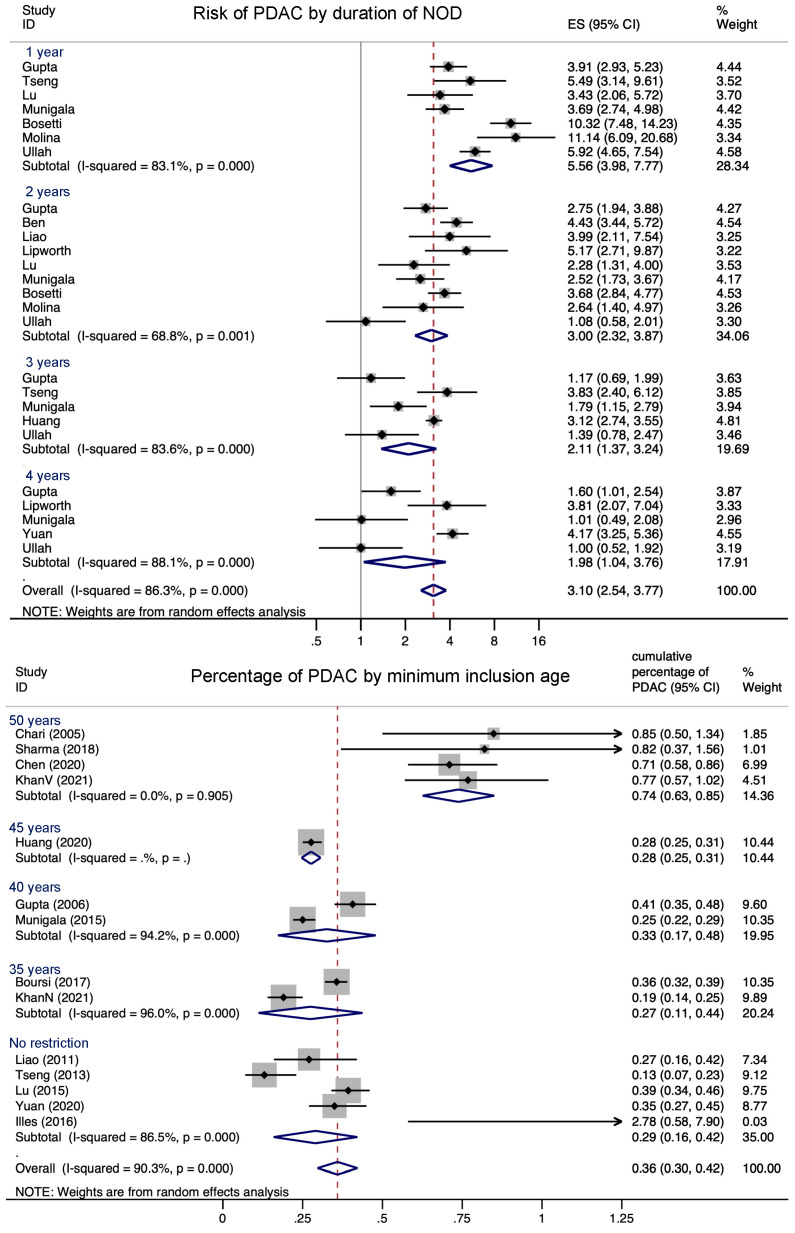
Meta analysis of OR of PDAC in NOD as opposed to no diabetes in patients grouped by the allowed duration of NOD as defined in the study or by the corresponding subgroup. The proportion of NOD with pancreatic adenocarcinoma as a probable reason for diabetes in the cohort studies in subgroups of applied age restriction. When only NOD older than 50 were included, it was highest.

**Table 1 cancers-14-04684-t001:** Studies, designs, and populations (References in brackets).

Author, Journal, Year	City or Region, Country, Database Name (When Available), Period (Years)	Study Design, Study Population, Sampling Method	Patient Characteristics in NOD (Mean Age, Obesity, Smoking)	NOD Definition
Gupta et al., Clin Gastroenterol Hepatol, 2006 [20]	USA, VA National Patient Care database 1998–2004	Retrospective cohort, veterans, all without previous diagnosis of PDAC or DM were included, 36,631 developed NOD, of which 149 had PDAC	US veterans > 40 years, in NOD cohort 97% male, average age 64 years	1 year
Boursi et al., Gastroenterology, 2017 [21]	UK, THIN database 1995–2013	Retrospective cohort, all patients with incident DM were included: 109,385 patients with NOD, of which 390 had PDAC	All < 35 years were excluded	<3 years
Lee et al., Journal of Clinical Gastroenterology 2012 [22]	Seoul, Korea, 2003–2009	Retrospective case-control Cases: 151 NOD with PDAC, Controls: 302 NOD, no cancer 1:2 matched, randomly selected	Mean age 61 years (cases) and 56 years (controls) 58% male in cases, 66% in controls	<2 years
Ben et al., European Journal of Cancer 2011 [23]	Shanghai, China, Hospital Data 2000–2009	prospective case-control Cases: 1458 PDAC, of which 307 NOD Controls: 1:1 matched for time of admission, age, sex, sociodemographic variables, 1528 of which 88 NOD	Mean age 62 years 67% male	<2 years
Liao et al., Journal of Gastroenterology and Hepatology 2012 [24]	Taiwan, National Health Insurance 1998–2007	Retrospective cohort, entire population, nested case-control: Cases: all DM, of which 6911 had NOD, and 19 PDAC Controls: No DM, 1:4 matched for age and sex, randomly selected	Mean age of 55.9 years 54% male, Obesity 2.43%	<2 years
Tseng et al., Pancreas 2013 [25]	Taiwan, National Health Insurance 2005–2006	Retrospective cohort, general population, random sample including 29,236 NOD and of those 32 PDAC	48.5% male	Groups 1, 3 or >3 years
Lipworth et al., Diabetes/Metabolism Research and Reviews 2011 [26]	Milan, Italy 1983–1992; 1991–2008	Combined data from two prospective case-control studies, hospital population, convenience sample, including 51 PDAC/NOD cases and 39 NOD controls	Median age 55 years (controls), 63 years (cases) 63% resp. 53% male	Subgroup < 2 years
Lu et al., British Journal of Cancer 2015 [27]	UK, THIN Database 1996–2010	Two retrospective cohorts from the general population NOD cohort 44,373, of which 175 had PDAC Control-cohort: 188,734 had no diabetes, of which 354 had PDAC	Mean age ~70 years (age groups) 58% male, 35% obesity, 23% current smokers	Groups 1, 2, 5, and >5 years
Müller et al., Pancreatology 2019 [28]	Great Britain, Clinical Practice Research Datalink (CPRD) 2004–2013	retrospective case-control Cases: 588 PDAC and NOD Controls: 5486 NOD, 1:10, matched for age, sex, time DM diagnosis, follow up	Mean age ~70 years (age groups), 49.5% male, 28.7% BMI > 30, 18% current smokers	<2 years
Munigala et al., Clinical and Translational Gastroenterology 2015 [29]	St Louis, USA, Veterans’ Health Administration national medical care data sets 1998–2007	Retrospective cohort, veterans, all without previous diagnosis of PDAC or DM, were included. 73,811 developed NOD, of which 183 had PDAC	Mean age 60.2 years, all < 40 years excluded by design 94% male, 74% white 46.8% obesity, 57% smoking	Groups 1, 2, 3, 4 years
Yuan et al., JAMA Oncology, 2020 [30]	USA Nurses’ Health Study (NHS), baseline 1978, Health Professionals Follow-Up Study (HPFS), baseline 1988	Two retrospective cohorts, female nurses and male physicians, without previous diagnosis of PDAC or DM. Within the patients with NOD, 67 PDAC cases were observed.	Mean age 69 years White 93.3%, Black 3.5% Obesity 43% Ever-smokers 56%	<4 years
Bosetti et al., Annals of Oncology 2014 [31]	International, USA, Canada, Greece, Central Europe, Italy, Australia, 1983–2012	Combined data from 15 case-control studies Cases: PDAC Controls: hospital/hospital visitors/populationNOD subgroup; including 525 NOD/PDAC cases	Not published for NOD subgroup	Groups < 1 years, 1–2, 2–5, >5
Illés et al., Pancreatology, 2016 [19]	Szeged, Hungary 2012–2014	Prospectively recruited, 108 patients with NOD, of which 3 had PDAC	Mean age 58 years 42.6% male, mean BMI 30.5, 29% ever smoker	<3 years
Chari et al., Gastroenterology 2005 [32]	Rochester, USA 1950–1994	Cohort of 2122 NOD including 18 PDAC with nested case-control: Cases: NOD with PDAC, 18 cases Controls: NOD, 1:4 matched for age, sex, time of diabetes diagnosis, 72 controls	All < 50 years excluded by design No demographic data on cohort	<3 years
Hart et al., Pancreas 2011 [33]	Rochester, USA 1981–2007	Retrospective case-control, 29 Cases: all NOD and PDAC in a 120-mile radius of Rochester 43 Controls: NOD matched for sex and age	Mean age 76 years cases, 72 years controls, 37% male cases, 56% controls	<3 years
Huang et al., Clinical Gastroenterology and Hepatology, 2020 [34]	Kaiser Permanente Southern California, USA (KPSC, Insurance) 2006–2016	Retrospective cohorts, all with sufficient data and without previous diagnosis of PDAC, were included. 110,699 NOD, of which 306 with PDAC	All < 45 years were excludedMean age 59 years, Male 52% Whites (44%), Hispanics (37%), Asians (15%) Blacks (15%).	<3 years
Sharma et al., Gastroenterology 2018 [35]	Rochester, USA, Rochester Epidemiology Project (REP) 2000–2015	Retrospectively collected data from 4 independent cohorts, with 64 PDAC/NOD and 192 NOD-Controls in the discovery set, and a cohort of 1096 NOD, including 9 PDAC in the validation set	All < 50 years were excluded Mean age 65.6 years, 50% male	<3 years
Chen et al., Digestive Diseases and Sciences 2021 [36]	Kaiser Permanente Southern California, USA (KPSC, Insurance) 2010–2014	Retrospective cohort of all patients without previous diagnosis of PDAC, meeting NOD criteria during the enrolment period, 13,947 NOD including 99 PDAC	All < 50 years were excluded No PDAC: 64.1 years, 48% male, 91 kg, PDAC: 69.2 years, 57% male, 84.4 kg	<3 years
Molina-Montes et al., Gut, 2021 [37]	PanGenEU, Europe, 28 centers from Spain, Italy, UK, Ireland, Germany, Sweden 2007–2014	Retrospective case-control, we used only data from the subgroup with NOD, with general population as control. Data on long-standing diabetes was ignored. It included 200 cases of PDAC/NOD	63.4% male, mean age ~65 years (age groups), 30.5% obese	<2 years
Khan, Pancreatology, 2021 [38]	TrinetX—Validation of ENDPAC	Retrospective cohort of 15,539 NOD patients, of which 48 had PDAC	<50 years excluded by design PDAC 68 years, 54% male, 81% white, 39% smokers No PDAC, 67 years, 50% male, 76% white, 21% smokers	<3 years
Khan, Pancreas, 2021 [39]	TrinetX—validation of Boursi	Retrospective cohort of 27,893 NOD patients, of which 52 had PDAC	<35 years excluded by design PDAC 74 years, No PDAC 64 years	<3 years
Ullah, BMC Cancer, 2021 [40]	EL-PaC-Epidem London, UK 2008–2020	Case-Control study, 965 PDAC, 3963 Non-malignant pancreatic disease, 4355 Controls	Mean age 55.1, 51% male, 54.4% white	Groups 1,2,3 years

## Data Availability

For access to data, contact the corresponding author.

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
