# Peer review of "Risk Factors for Pancreatic Cancer in Patients with New-Onset Diabetes: A Systematic Review and Meta-Analysis"

_cancers, 2022, doi:10.3390/cancers14194684_

Round 1

Reviewer 1 Report

The association of diabetes and pancreatic cancer is a long-standing controversial research topic.   This meta-analysis attempted to describe risk factors for PDAC in patients with new-onset diabetes (NOD).  In the introduction, it correctly stated the problem that diabetes, especially NOD, could be a consequence rather than a cause of PDAC. The clinical challenge is how to identify NOD patients with an occult PDAC.  Thus, describe risk factors and clinical features of NOD-associated PADC have potential values in early diagnosis of PDAC. 

The major concern on this study is the confusing study design.  It is not clear what is compared or what the definition of case and control is.   In the 22 studies included in this meta-analysis, some studies focused on PDAC patients with NOD, while some studies included all PDAC cases regardless their diabetes status (e.g. Ben 2010; Tseng 2013; Lipworth 2011; Bosetti 2014 etc).  If the cases were selected from these studies by diabetes duration, why there are so many such studies (e.g. references 13, 51, 69, 71, 87, 92, 102, etc.) were excluded by the criterion of “wrong study population”? Some studies are excluded for “wrong outcome” even though the study’s end point is PDAC.  The total number of study subjects were 576,210 patients with NOD, what is the number of PDAC cases? Because PDAC is a relatively rare cancer, the study power is often limited by the small number of cases available.  It is necessary to report the number of cases involved.

 It is even more confusing what controls were selected for this analysis.   Were all the controls NOD patients or a mixture of patients with or without diabetes? In the abstract, it was stated that the objective of the study was to “Examine the risk for PDAC in NOD patients compared with persons without diabetes”, this is a confusing statement. No data is presented to echo this claim in the text.   Even though the authors stated that they performed subgroup analyses by diabetes status, no details on the method of stratification and the final finding was presented in the results. As pointed out by the authors, patient selection is the most crucial factor for overall risk of bias and variability of patient definitions influenced the results considerably. Then the authors really need to clearly describe the selection criteria for their cases and controls from which the risk was estimated.

The reference citing in the article was sloppy with many mistakes.  For example, reference “Boursi et al. Gastroenterology” seems to have a wrong publication year, it should be 2017 not 2013.  Reference 16 and 86: “Gastroenterology” was misspelled as “Gstroenterology”. Reference 36 can’t be found in the literature according to the information provided in the Table.  

The 22 studies included in the analysis were cited in the text sometimes by numbers and sometimes by names.  The last three references on the reference list were not cited in the text and they seem to be studies included in the meta-analysis.  Please check your consistency. 

Reference 53 in Table A1, Chari, 2008, Gastroenterology, Pancreatic cancer–associated diabetes mellitus: prevalence and temporal association with diagnosis of cancer; and reference 86 Pannala, 2008 Gstroenterology, Clinical profile of pancreatic cancer-associated diabetes mellitus.  Those are landmark studies on PDAC and NOD, both are excluded for “Wrong study population”.  Why?

It is weird that some well-designed case-control studies were excluded for “no risk factor”.   

The lack of association of obesity and PDAC in this analysis is mostly likely due to weight loss in PDAC patients.  This should be considered in the discussion.

Minor comments:

BMI was dichotomized at 30, i.e. >30 vs < 30, what about BMI=30?

Reviewer 2 Report

This manuscript well summarized current evidences about early detection of pancreatic cancer based on new-onset diabetes. Included studies are comprehensive, and conclusion supports previous reports in this field. This manuscript seems appropriate for publication in its present form. I recommend publication. 

Author Response

Dear Reviewer,

We thank you very much for your work, and are very glad that you find our paper appropriate for publication. 

Round 2

Reviewer 1 Report

The authors addressed my previous concerns to my satisfaction.  I have some minor comments on Table 1.

1. The number of reference citation should be explained on the title row. 

2. References 38 and 39 are labeled with "V" and "N", please clarify what they represent. 

3. Reference 34, the number 11,0393 needs to be corrected. 
